# Reducing the Effectiveness of Ward Particulate Matter, Bacteria and Influenza Virus by Combining Two Complementary Air Purifiers

**DOI:** 10.3390/ijerph191610446

**Published:** 2022-08-22

**Authors:** Bingliang Zhou, Tiantian Liu, Siqi Yi, Yuanyuan Huang, Yubing Guo, Si Huang, Chengxing Zhou, Rong Zhou, Hong Cao

**Affiliations:** 1Guangdong Provincial Key Laboratory of Tropical Disease Research, Department of Microbiology, School of Public Health, Southern Medical University, Guangzhou 510515, China; 2School of Public Health, Guangdong Pharmaceutical University, Guangzhou 510310, China; 3State Key Laboratory of Respiratory Disease, National Clinical Research Center for Respiratory Disease, Guangzhou Institute of Respiratory Health, First Affiliated Hospital of Guangzhou Medical University, Guangzhou Medical University, Guangzhou 510180, China

**Keywords:** combined application, air purifier, purification rate, particulate matter, *Staphylococcus albus*, influenza virus

## Abstract

Air purifiers should pay much attention to hospital-associated infections, but the role of a single air purifier is limited. The goal of this study was to evaluate the effectiveness of the combined application of the nonequilibrium positive and negative oxygen ion purifier (PNOI) and the high-efficiency particulate air filter (HEPA) on a complex, polluted environment. Two of the better performing purifiers were selected before the study. The efficacy of their use alone and in combination for purification of cigarette particulate matter (PM), Staphylococcus albicans, and influenza virus were then evaluated under a simulated contaminated ward. PNAI and HEPA alone are deficient. However, when they were combined, they achieved 98.44%, 99.75%, and 100% 30 min purification rates for cigarette PM, *S. albus*, and influenza virus, respectively. The purification of pollution of various particle sizes and positions was optimized and reduced differentials, and a subset of airborne influenza viruses is inactivated. Furthermore, they were superior to ultraviolet disinfection for microbial purification in air. This work demonstrates the strong purification capability of the combined application of these two air purifiers for complex air pollution, which provides a new idea for infection control in medical institutions.

## 1. Introduction

Since 2019, the world has remained torn by the novel coronavirus, and it is undeniable that it allows us to recognize that hospital-associated infection (HAI) control remains inadequate in healthcare settings today. It the early stage of the COVID-19 epidemic, among 138 hospitalized patients with COVID-19 in a hospital in Wuhan, China, 41% of them were suspected to be related to hospital transmission [1]. In a major London teaching hospital, 66/435 (15%) of COVID-19 inpatient cases between 2 March and 12 April 2020 were definitely or probably hospital-acquired through varied transmission routes [2]. Between March 2020 and February 2021, 185 (8.6%) cases were considered cases of nosocomial transmission in a tertiary university hospital in the city of Sao Paulo, Brazil [3]. Of course, the reality is far more than these, exposing the severity of HAI as early as 2002 with severe acute respiratory syndrome coronavirus (SARS-CoV) [4] and in 2012 with Middle East respiratory syndrome coronavirus (MERS-CoV) [5]. Nearly 1.7 million hospitalized patients in the United States are simultaneously infected with HAI every year, of which more than 98,000 patients die of HAI [6]. Clearly, the hazards of HAI were identified many years ago, but the control of HAI remains a difficult problem in current healthcare systems [7].

Due to the particularities of the hospital, its environment contains a large number of microorganisms [8], and it provides very favorable conditions for the reproduction and spread of pathogenic microorganisms [9,10]. On the other hand, as a major site of antibiotic use, HAI and antimicrobial resistance mutually promote the formation of a vicious cycle [11,12]. There is strong evidence that airborne and aerosol transmission of pathogenic microorganisms are widespread in hospital environments [13]. In addition to this, several meteorological factors influence the survival and spread of nosocomial environmental pathogens, such as inappropriate air filtration and recirculation that can exacerbate virus spread on wards and other adjoining spaces (i.e., corridors) [14]. The higher relative humidity may lengthen the survival time of the new corona virus in the air [15]. The increase in temperature facilitates the multiplication of microorganisms in the ward environment [16], and patients and medical staff act as vectors of infection between medical institutions [17]. This means that, without protection, the nosocomial population is consistently exposed to a multiple infection risk [18].

To improve HAI status, traditional views have focused on strategies to eliminate pathogens present on patients, clinical surfaces, and health care workers [19], which has prompted the implementation of various infection control and disinfection protocols that have also been successful in reducing the incidence of HAI [20]. However, recent studies have shown that a significant proportion of pollutants causing infections in the human population are airborne [13,21]. Improving health care expenditure and reducing hospital air pollution can effectively reduce the mortality rate of SARS-CoV-2 [22], and mechanical air conditioning and natural ventilation technologies have a positive effect on hospital air purification [23]. Therefore, the application of various air purification technologies in hospitals has good prospects. Some studies have evaluated the HEPA cleaner and found that it reduces the average room PM_2.5_ concentration from 17.0 μg/m^3^ to 9.26 μg/m^3^ and reduces the medication burden in asthmatic children [24]. More studies have shown that some air purification technologies are effective for a certain contaminant; for example, HEPA purifiers reduced the concentration of phiX174 bacteriophages in aerosol by 99.9974–99.9999% [25], HUCoV-229E was inactivated in less than 60 min on brass nanomaterials containing at least 70% copper [26], and so on. Visible air cleaners may have an effect on a certain contaminant in room air but are unknown for hospital wards where multiple contaminants are present. A rigorous and feasible line of research is needed in the area of air filtration and recirculation in healthcare facilities, ensuring the ability to respond to possible new outbreaks [14].

To ask the purifier scheme that can cope with multiple pollutants, two air purifiers based on HEPA and nonequilibrium positive and negative oxygen ions were selected after testing multiple air purification units in this study. We evaluated the decontamination capacity and deficiencies of their separate use and explored the combined use on different pollutants in indoor air. The study created heavily polluted environments of cigarette PM, *S. albus*, and influenza virus, and their concentrations before and after purification were detected to evaluate differences in indoor air pollutants by position, particle size, and decontamination efficacy. In addition, the findings can provide a reference for the effective application of air purification equipment in hospital wards.

## 2. Materials and Methods

### 2.1. Pollutant and Experimental Equipment

*S. albus* standard strain 8032 (Guangdong Institute of Microbiology, Guangzhou, China), cigarettes (Hongta Mountain, Yunnan, China) and H1N1-pr8 influenza virus (obtained from internal stock) were selected as simulated indoor air pollutants in this study. *S. albus* was cultured in agar medium (nutrient agar, HuanKai Microbial, Guangzhou, China). Influenza virus was inoculated and amplified in 10-day-old embryonated chicken eggs (emerging Huanong, Guangzhou, China). MDCK cells were cultured in Dulbecco’s Modified Eagle medium (containing 100 U/mL penicillin, 100 μg/mL streptomycin and 10% fetal bovine serum, Shanghai Excell Bio, Shanghai, China). An ABI7500 real-time PCR machine determined the number of virus copies (Thermo Fisher Science, Waltham, MA, USA).

Viral copy number was determined using an ABI7500 real-time PCR machine (Thermo Fisher Science, Waltham, MA, USA). Aerosols were generated using microbial aerosol generator (Kangjie Instrument Research Institute, Liaoyang, China). Particles were determined by Y09-301 laser dust particle counter (WeiTian Environmental Technology Co., Ltd., Suzhou, China). Bacterial aerosols were collected with FA-1 six-stage sieve hole percussive air microbial sampler (Jintan District Jincheng Shuohua instrument Factory, Figure 1), and influenza virus was collected with ZW-100 portable large flow bioaerosol sampler (Guangzhou medium and Micro Technology Co., Ltd., Guangzhou, China).

### 2.2. Experimental Scene

We adapted a container to model a standard ward (Figure 2). The room had an effective area of 16.0 m^2^ and a volume of 41.6 m^3^. It was placed on a ground platform with less staff movement, was closed, and was less affected from the outside world. The in-house included a regular ward facility, including a bed, table, chair, locker, bathroom, washing equipment, etc. As the basic living environment for patients, the basic structure of this ward meets the standards for environmental science (Table 1). According to the sampling criteria and the indoor arrangement, we selected 7 sampling sites (A–G) in the room.

### 2.3. Air Purifiers

Before conducting formal experiments, we tested five kinds of air purifiers, namely, HEPA purifier, nonequilibrium positive and negative oxygen ion purifier (PNOI), negative air ion generator, photocatalyst purifier, and neowind purifier. After comprehensive analysis, the best performing HEPA and PNOI were selected. HEPA purifier uses a filter system composed of HEPA, and air contaminants in the filter are constantly blocked from the filter when it reaches between the filter trap and the fiber through a 0.024 m^2^ vent at 1.0 m/s speed. The manufacturer declared its ability to reduce PM_2.5_ below 35 μg/m^3^ and to ≥99.0% purification of *S. albus* in 1 h. PNOI, which produces nonequilibrium positive and negative oxygen ions at a lower voltage, can kill planktonic bacteria in the air by collision and the redox effect of positive and negative ions, and its vents have an effective area of 0.88 m^2^ and wind speed of 2.5 m/s. The assay reports that it is capable of killing≥ 99.8% of *S. albus* in 1h. They are currently the more mainstream air purification units and are less expensive and easy to install.

### 2.4. Formation of Heavily Polluted Environment

Within the infectious disease ward, the contaminant was mainly generated by the patients while the patients were mostly bed-ridden, so we took the central C of the bed (Figure 2) as the point of occurrence for the contaminant. We used PBS to dissolve the *S. albus* cultured for 24 h to make the absorbance of 0.48–0.51 and diluted it another 1000 times for use. The influenza virus with the PCR assay result of CT = 15–20 was selected for use. The prepared bacterial or viral fluid was added into the microbial aerosol generator and sprayed for 15 min, and the air was mixed thoroughly with a fan to form a room environment heavily contaminated with bacteria or viruses. The cigarette was directly ignited at C for 15 min, and then a heavily polluted environment with PM was formed after mixing.

Additionally, the microbial aerosol generator was continuously turned on until the end of sampling after the heavily polluted environment was formed, thus creating a continuously contaminated ward environment.

### 2.5. Collection and Detection of Samples

The samples for the first round were collected at each sampling site (0.5 m from the wall surface and 1.4 m from the ground surface) after the environment was stabilized. After acquisition was completed, samples were collected once each at 10, 20, and 30 min with the purifier turned on, and sampling without turning on the purifier served as a control. The bacterial samples were incubated 24 h in a bacterial incubator at a temperature of 37 °C before counting. Viral samples RNA was partially extracted and viral copy number was determined using a real-time PCR. MDCK cells were infected with the positive viral samples, and cytopathic changes were observed using an inverted microscope after one day of culture. The amount of PM can be read on-site, and the analysis can be performed when the results are saved.

### 2.6. Quality Control

Prior to formal experiments, we performed multiple pre-experiments with rigorous training and division of labor for experimenters. Clean clothes, hats, and shoe covers were worn during sampling to ensure that the operation of the experiment was smooth. On the other hand, UV disinfection was performed for 30 min before the experiment, and the closed standing was allowed to stand for 3 h, and a convenient dehumidifier and air conditioning were used to regulate the temperature (20–28 °C) and humidity (50–70%) in the room to ensure that the heat and humidity were relatively stable.

### 2.7. Data Analysis and Calculation Formula

We calculated acquired pollutant concentrations using Excel (Microsoft Office Professional Enhanced Edition 2019, Microsoft, Redmond, WA, USA) and summarized categorically. Means and standard deviations were calculated with IBM SPSS statistics 26, and *t* tests and ANOVA were performed to evaluate statistical differences among variables (ns = no significant difference, * *p* < 0.05, ** *p* < 0.01, *** *p* < 0.001, α = 0.05). Plotting of graphs was performed using GraphPad prism 8.

Calculation formula for pollutant purification rate is as follows:K = (C_0_ − C_t_)/C_0_

K: purification rate; C_0_: Initial pollutant concentration; C_t_: t minute pollutant concentration.

## 3. Results

### 3.1. Preliminary Purification of Three Air Pollutants by Two Purifiers

Before testing the effect of the combination, we individually evaluated the efficacy of HEPA and PNOI to purify airborne PM, *S. albus*, and influenza virus. For environments heavily contaminated with PM (Figure 3), only a 55.89% average purification rate was observed after 30 min of the PNOI running, and the average PM_2.5_ and PM_10_ concentrations were 42.76 μg/m^3^ and 64.38 μg/m^3^, respectively. On the contrary, HEPA showed a good purification efficiency, achieving a 97.23% average purification rate at 30 min and PM_2.5_ and PM_10_ average concentrations of 5.48 μg/m^3^ and 14.05 μg/m^3^, respectively, but the purification rate was not sufficient (10 min: 52.00%). The purification rate of *S. albus* by HEPA reached 99% in 30 min, and the number of colonies was reduced to 464.41 cfu/m^3^. The purification effect of the PNAI running 10 min was significantly better than that of the HEPA (*p* < 0.001), but the purification at 30 min was not significantly different from that of the HEPA (*p* > 0.05). For influenza virus, the PNOI was globally leading the HEPA in purification capacity (*p* < 0.001 at each time point) and completely purified the virus by 30 min.

After that, we turned on the two purifiers simultaneously. The purification efficiency and speed were significantly improved (*p* < 0.001), with an 86.25% purification rate at 10 min, and the average PM_2.5_ and PM_10_ concentrations at 30 min were only 0.81 μg/m^3^ and 2.48 μg/m^3^, respectively, reaching very clean levels. For both bacteria and viruses, purification was accelerated over a short period of 10 min compared to PNOI (their purification rate at 10 min was comparable to that of a 20 min PNOI run); *S. albus* eventually decreased to 5.05 cfu/m^3^, and the influenza virus was completely cleared.

### 3.2. Combined Application was Effective for Different Positions and Particle Sizes

After detecting the size of each of the particles and aerosols at different sites, we further analyzed the effect of the air cleaner on complex spaces and particle sizes. After the particles were purified for 30 min, the PNOI against a diameter of 0.3–0.5 μm had a significantly poorer purification effect than the other particle sizes (*p* < 0.001). Interestingly, HEPA was relatively less effective for particles larger than 5.0 μm in diameter (*p* < 0.001). Predictably, the combined application of the two purifiers had a significantly better purification effect on each particle size than did their application alone, and only the particle purification rate (88.46%) from diameter 0.3–0.5 μm was slightly lower than that of the others (*p* < 0.05). Based on the characteristics of the bacterial sampler, we measured the number of colonies contained in aerosols in six particle size ranges. At 30 min operation, PNOI had a slightly lower effect on aerosols with a diameter of 0.65–2.1 μm than other particle sizes (*p* < 0.001), while the purification rates for the combined applications showed no significant difference in particle size (*p* > 0.05), and all had good effects (Table 2).

Analyzing the different positions (Table 3), HEPA showed a slightly lower PM decontamination rate in D (*p* < 0.01). PNOI did not differ significantly across sites (*p* > 0.05) but purification was general. After the combined application, the purification rate of D was slightly lower than that of the other positions, but they were not statistically different (*p* > 0.05); meanwhile, the purification effect was also significantly better than that of the single purifier (*p* < 0.001). Unlike PM, the bacterial purification at different positions by two purifiers applied individually or in combination was excellent and showed no significant difference (*p* > 0.05).

### 3.3. Combination Application Accelerates Influenza Virus Inactivation

Previous experiments used RT-PCR to test samples for influenza virus content, but viral activity could not be determined. We infected MDCK cells with virus-positive samples (Table 4) and found that both influenza viruses were active in the positive samples from HEPA (48/48), while the virus was still active in the samples from PNOI 57.50% (23/40). After the combined application, the virus activity decreased to 37.50% (12/32), of which none of the positive samples at 20 min showed activity.

### 3.4. Combination Application Better than Common UV Disinfection

UV disinfection is one of the best methods for indoor environment disinfection. To evaluate the difference in decontamination between common UV lamps and our purifiers, we examined the efficacy of ultraviolet lamps for *S. albus* and influenza virus using the same method (Figure 4). In contrast, it was found that there was no significant difference between bacterial purification by the combination of two purifiers and UV lamps (*p* > 0.05). For influenza virus, the average purification rate of UV disinfection was 93.43%, which could not completely purify the influenza virus in the room air at 30 min and was less effective than the purifiers combination (*p* < 0.05). In a general ward where UV lamps are inconveniently applied, it is obvious that the combination of air purifiers can replace UV lamps to play an effective air purification role.

### 3.5. Combination Applications Still Have Better Effects on Continuously Polluted Environments

The previous experiments evaluated the direct effect of the combination of the two cleaners and gave positive results, but because patients live for long periods of time in the hospital, the contamination of the ward is often persistent. We therefore evaluated their effects in a simulated continuously polluted environment (Figure 5). Obviously, both bacterial and viral purification effects of the continuously contaminated environment were slightly lower than those of the primary contaminated environment (*p* < 0.05), but at 30 min, the decontamination rates could both reach 90%. Their purification rates against *S. albus* continued to increase over time (*p* < 0.05), whereas those against influenza viruses stabilized at around 90%.

## 4. Discussion

This study tested the purification efficacy of two air purifiers for a heavily contaminated ward environment. The purification effect of HEPA filtration technology on indoor PM and aerosols is widely recognized [27,28,29]. PNOI is based on this emerging technology of nonequilibrium positive and negative oxygen ions, and our study demonstrates a superior capability in decontaminating bacteria and viruses, with the released ions also inactivating part of airborne influenza viruses. After multifaceted analysis, we confirmed that the two purifiers combined obviously strengthened their air purification capacity and were able to effectively purify complex room air pollution for a short period of time.

PM, as one of the most dominant pollutions in the air, i.e., fine particles with a diameter of ≤5 μm, persists in the air for a long time and can easily enter the lower respiratory tract [30,31]. The small particle is orders of magnitude larger in air, and when the purifier is used, it falls far more easily than the large particles, exhibiting relatively better particulate matter purification below 5 μm in diameter, which is consistent with Dubey’s findings [27]. In addition to PM, aerosols carrying pathogenic microorganisms are ubiquitous in ward air [9,32]. Agarwal’s study in New Delhi indicated 1.1–4.7 μm bioaerosols with more microorganisms [33]. It has also been shown that the majority of sporadic viral RNA from coughing in influenza patients is contained in particles in the respirable size range [34]. Having good effecst on fine PM and a smaller aerosol (diameter ≤ 5 μm) are important criteria that air purifiers need to achieve. Obviously, the purifier combination in this study was able to do this.

In this larger and separate space in hospital wards, one or more bacteria, and even super-resistant bacteria and viruses, can be detected in the exhaled gas of nearly half of hospitalized patients [35,36], which gradually spreads throughout the room. When the purifier was run alone in this study, it showed poor effects on some locations, which was greatly related to the purifier ventilatory efficiency and room layout. For example, in a ward with a patient with SARS-CoV-2, the virus was widely distributed on the floor, a computer mouse, trash buckets, and the patient’s bedside banister and was detected in the air about 4m away from the patient [37], even as the air flow reached the outside of the ward [38]. Interestingly, PNOI produces ions capable of diffusing everywhere with the aid of the ventilation system and the airflow of HEPA. As the present study presented results that bacteria and viruses were more easily purified, the purification rates were not statistically different among the seven sampling sites. Another unexpected phenomenon was that the pathogenic aerosols derived from patients or those carrying pathogenic bacteria are highly prone to stay in humid environments, such as those in toilet air, pools, and buckets [39], where microbial decontamination is relatively poor. Because the toilet is small, installing an efficient cleaner for the toilet is clearly cost-effective. A ward toilet was not actually used as simulated in this study, so decontamination may have been overestimated. Purifiers need to be considered for installation near more contaminated areas, such as toilets, in real-world situations or for enhanced sanitization of these areas.

At present, one of the most common and predominant methods of disinfection of areas such as hospital corridors, wards, nurse stations, and doctors’ offices is the use of disinfectants for the cleaning of surfaces on the ground and objects [40]. This approach, while fully effective, requires significant human effort to be invested and is not durable. Studies have shown that heating, ventilation, and air-conditioning systems in hospital wards are the main indoor ventilation facilities and another important means of infectious disease control, but their incorrect use can instead lead to the spread of disease [41]. Under the influence of the new corona virus, the air cleaner has a better prospect for use in the medical arena. However, as the HEPA purifier in this study was capable of clearing airborne particulates and aerosols by requiring air to pass through its strainer, it is clear that its effects are not comprehensive. There are studies evaluating the effectiveness of a certain plasma cleaner for 20 days, and it was found that it did not affect bacteria and fungi in the air of the hospital ward [42]. Marc’s study indicated that no statistically significant differences were found between the use of portable air disinfection systems and the use of natural HVAC systems in two health care self-learning rooms [43]. Therefore, in complex air pollution locations such as hospitals, proper selection of air purification equipment is required. The combined application schemes demonstrated in this study do exhibit good synthetic ability, but this does not mean that they are the best combination. This study hopes to give more ideas and inspiration to related researchers and producers while providing a protocol for the use of a purifier and also to demonstrate an effective method for the comprehensive evaluation of purifiers.

In summary, under a simulated heavily polluted environment, the combination of two kinds of purifiers with different characteristics effectively improved indoor air particulate and microbial pollution and were also fast and effective. However, failure to evaluate a real scenario is a limitation of this study. Further validation of their actual effects in real-life scenarios is also required.

## 5. Conclusions

In this study, we comprehensively evaluated the effects of two air purifiers, alone and in combination, by simulating the indoor environments with severe PM, *S. albus*, and influenza virus pollution. The results showed that HEPA and PNOI alone had limited effectiveness for purification. However, in combination, it can obviously improve the purification capacity, narrow the purification difference for pollutants in different positions and for different particle sizes and inactivate some airborne influenza viruses. Additionally, it has a better purification effect on airborne microorganisms than ordinary UV disinfection.

## Figures and Tables

**Figure 1 ijerph-19-10446-f001:**
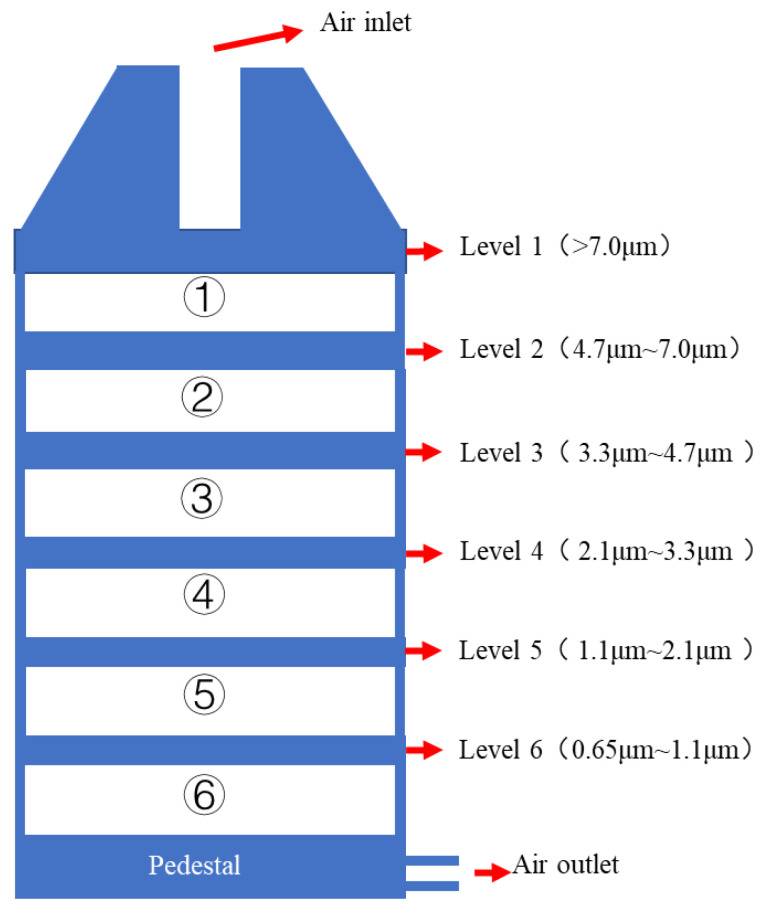
Structure diagram of bacterial sampler. ①–⑥ are the places where agar plates are placed, and levels 1–6 can selectively collect aerosols with corresponding particle size.

**Figure 2 ijerph-19-10446-f002:**
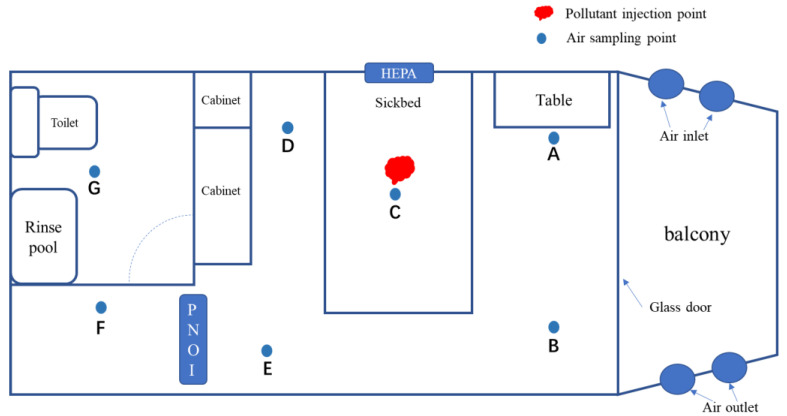
It contains the infrastructure of the standard ward (shown in part of the figure), pollutant injection points, and 7 (A to G) sampling positions. In addition, HEPA is installed 2 m above the head of the bed, and PNOI is installed at the central air conditioning outlet on the right side of E.

**Figure 3 ijerph-19-10446-f003:**
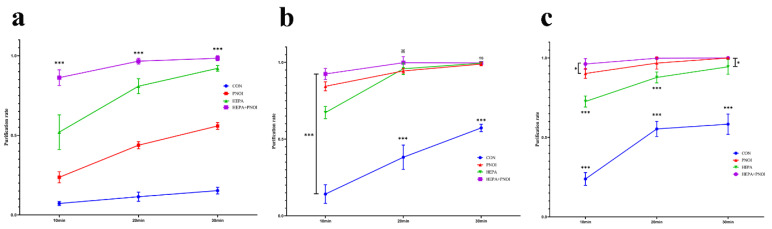
Purification rate of air pollutants with each module. (**a**) Cigarette particlulate matter; (**b**) *S. albus*; (**c**) influenza virus. The statistical difference for the last time of different modules is analyzed (ns: no significance; *: *p* < 0.05; ***: *p* < 0.001).

**Figure 4 ijerph-19-10446-f004:**
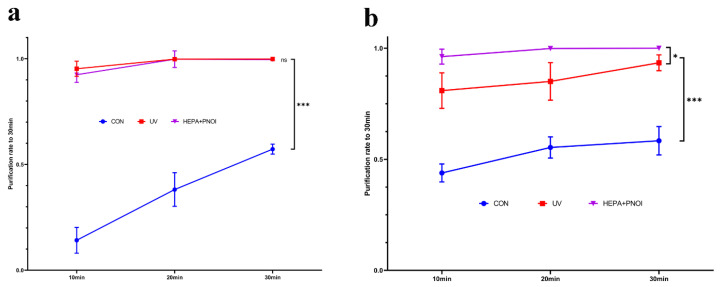
Comparison of the efficacy of purifier combination and UV lamps in removing *S. albus* (**a**) and influenza virus (**b**). The statistical differences between the items are analyzed. (ns: no significance; *: *p* < 0.05; ***: *p* < 0.001).

**Figure 5 ijerph-19-10446-f005:**
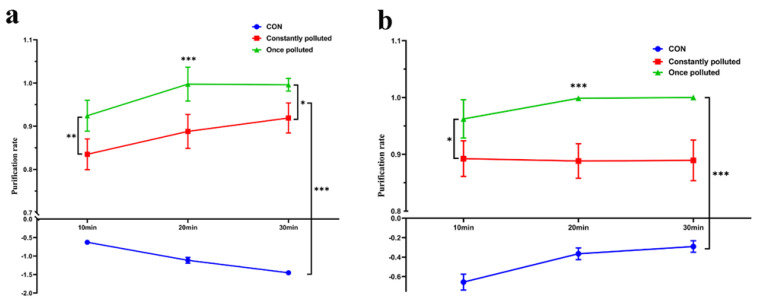
Comparison of the effects of combined application on the removal of *S. albus* (**a**) and influenza virus (**b**) from constantly and once-polluted environments. The statistical differences between the items are analyzed. (*: *p* < 0.05; **: *p* < 0.01; ***: *p* < 0.001).

**Table 1 ijerph-19-10446-t001:** Index of indoor environment in simulation ward.

Index	Actual Level
Room volume	41.62 m^3^/P
Net story height	2.61 m
Coefficient of room depth	2.21
Daylight factor	≥1.0%
Total viable count	<500 cfu/m^3^
Fresh air volume	30–60 m^3^/(h·P)
PM_10_	<0.05 mg/m^3^
Temperature	18–28 °C
Humidity	50–80%

**Table 2 ijerph-19-10446-t002:** Six particle sizes of particles and aerosol purification rate (%, mean (SD)).

Pollutant	Diameter (μm)	CON	HEPA	PNOI	HEPA + PNOI
PM	0.3–0.5	6.20 (1.41)	88.63 (2.71)	41.68 (2.55)	97.98 (0.22)
	0.5–1.0	21.67 (2.18)	94.40 (1.33)	64.88 (1.87)	98.57 (0.74)
	1.0–3.0	53.45 (1.97)	97.52 (0.52)	84.50 (0.63)	96.86 (0.52)
	3.0–5.0	7.08 (6.11)	82.74 (3.63)	65.31 (2.76)	88.46 (3.16)
	5.0–10.0	14.46 (9.76)	75.75 (4.91)	67.75 (2.86)	93.76 (2.62)
	>10.0	32.80 (7.07)	67.73 (4.22)	81.15 (2.75)	98.16 (2.93)
Bacterial aerosol	0.65–1.1	46.59 (8.76)	99.52 (0.45)	97.79 (0.70)	99.85 (0.19)
	1.1–2.1	20.97 (5.65)	99.54 (0.31)	96.92 (0.80)	99.85 (0.15)
	2.1–3.3	42.69 (3.42)	99.64 (0.26)	99.50 (0.28)	99.91 (0.11)
	3.3–4.7	72.17 (9.94)	99.52 (0.33)	99.88 (0.16)	99.93 (0.11)
	4.7–7.0	85.81 (4.05)	99.33 (0.61)	99.64 (0.42)	99.47 (0.50)
	>7.0	75.30 (6.83)	99.21 (0.75)	99.50 (0.79)	98.59 (0.28)

**Table 3 ijerph-19-10446-t003:** PM and aerosol purification rates at 7 positions (%, mean (SD)).

Pollutant	Position	CON	HEPA	PNOI	HEPA + PNOI
PM	A	30.10 (7.30)	86.78 (8.18)	74.32 (6.03)	91.45 (7.84)
	B	32.06 (8.33)	84.82 (6.82)	74.91 (6.09)	95.73 (2.92)
	C	29.26 (5.24)	91.24 (6.85)	73.48 (6.06)	93.13 (5.99)
	D	25.44 (5.07)	79.13 (7.51)	76.37 (6.28)	88.83 (9.38)
	E	26.19 (6.24)	88.86 (9.09)	69.12 (6.12)	95.62 (5.43)
	F	30.32 (6.23)	83.42 (5.35)	67.42 (6.40)	92.04 (6.97)
	G	19.07 (8.59)	87.51 (6.27)	72.23 (6.59)	95.35 (5.04)
Bacterial aerosol	A	58.04 (4.58)	99.20 (0.50)	98.55 (0.75)	99.92 (0.12)
	B	48.98 (5.10)	99.47 (0.44)	98.45 (0.4)	99.62 (0.48)
	C	60.42 (7.68)	99.30 (0.62)	99.19 (0.71)	99.62 (0.64)
	D	58.27 (9.15)	99.58 (0.35)	99.21 (0.63)	99.86 (0.32)
	E	63.50 (5.90)	99.49 (0.33)	98.82 (0.75)	99.75 (0.32)
	F	50.39 (5.08)	99.86 (0.16)	99.12 (0.70)	100.00 (0)
	G	61.13 (5.50)	99.12 (0.59)	98.73 (0.73)	98.43 (0.29)

**Table 4 ijerph-19-10446-t004:** Result of influenza virus activity assays.

Purifier	Active	Inactive	Total	Activity Ratio (%)
CON	67	0	67	100.00
HEPA	48	0	48	100.00
PNOI	23	17	40	57.50
HEPA + PNOI	12	20	32	37.50

## Data Availability

Data sharing is not applicable. No new data were created or analyzed in this study.

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
