# Peer review of "Reducing the Effectiveness of Ward Particulate Matter, Bacteria and Influenza Virus by Combining Two Complementary Air Purifiers"

_ijerph, 2022, doi:10.3390/ijerph191610446_

Round 1

Reviewer 1 Report

Hi

Dear prof.

Reducing the effectiveness of indoor particulate matter, bacteria and influenza virus by combining two complementary air purifiers

The topics of this paper are interesting, but the structure and content must be revised by authors before to be reconsidered for publication.

In abstract

When applied aloneboth PNOI and HEPA have deficiencies. When these were combined, the 30 min purification rates for cigarette PM, S. albus, and influenza virus were 98.44%, 99.75%, and 100%, respectively” please mixed in one clear sentencesin

 In troduction

Please added about some climate factors e.g., wind speed, wind direction, temperature and humidity and..

Please used of some paper:

SARS-CoV-2 detection in hospital indoor environments, NW Iran https://doi.org/10.1016/j.apr.2022.101511
Mousavi, E.S., Kananizadeh, N., Martinello, R.A., Sherman, J.D. 2021COVID-19 Outbreak and Hospital Air Quality: A Systematic Review of Evidence on Air Filtration and Recirculation,Environmental Science and Technology55(7), pp. 4134-4147

Coccia M. 2021. High health expenditures and low exposure of population to air pollution as critical factors that can reduce fatality rate in COVID-19 pandemic crisis: a global analysis. Environmental Research, vol. 199, Article number 111339, https://doi.org/10.1016/j.envres.2021.111339

Kenarkoohi, A., Noorimotlagh, Z., Falahi, S., (...), Pakzad, I., Bastani, E.    2020, Investigation of hospital indoor air quality for the presence of SARS-Cov-2, Science 748,141324

In methodolody

Please written about QA/QC

Author Response

请参阅附件。

Reviewer 2 Report

On the one hand, the article is interesting (air purification is an important topic in many "indoor" environments), while on the other hand, even without conducting research, it can be said with high probability that after using two purifiers the total effect should not be worse than the effect after using one of the air purifiers (from the set of analyzed air purifiers). The only question is, how much better effect we will get by using two devices instead of one, and the authors of the article wanted to answer this question.

Here are some detailed comments:

- The title is inadequate to the content of the article - in the title is "indoor", while the article is mainly about hospitals. If the article is going to be about general "indoor" (hospitals/schools/offices, ...) then the introduction should be expanded, otherwise the title must be changed.

lines 64 - 71. maybe some details on how much other researchers managed to clean the air? And what will the research presented in the article be better at? Have two purifiers been tested at once in other researches?

line 84 - twice "were selected"

line 93 - "Particles" instead of "particles"

subsection 2.3 - what is the air cleaning efficiency for the presented purifiers (e.g. declared by the manufacturer)?

line 144 "The" instead of "the"

lines 167 and later - upper index for "3" in "m3"

Section "Discussion" - This is essentially a discussion of the benefits of clean air, not of research results. Of course, the result of the research was that the two air purifiers were effective in cleaning the air. However, it seems to me that this type of discussion is more suitable for the "Introduction" section than for the "Discussion"

I do not understand one more, and the description of the experiment does not clarify it - was the effectiveness of the purifiers tested after the polluted environment was created, then the sources of contamination (including viruses) were turned off and the effectiveness of the cleaning devices was tested? In a real situation, there would still be sources of contamination (viruses) in the form of patients, so the question arises, how effectively would both devices work under real conditions (with a continuous source of viruses / bacteria - i.e. patient / patients)?

Round 2

Reviewer 2 Report

Thank you for all changes.